# Implication of tropical lower stratospheric cooling in recent trends in tropical circulation and deep convective activity

**Kunihiko Kodera[1], Nawo Eguchi[2], Rei Ueyama[3], Yuhji Kuroda[1], Chiaki Kobayashi[1], Beatriz M. Funatsu[4], Chantal Claud[5], and Leonhard Pfister[3]**

[1] Meteorological Research Institute, Tsukuba, 305-0052, Japan
[2] Research Institute for Applied Mechanics, Kyushu University, Kasuga, 816-8580, Japan
[3] NASA Ames Research Center, Moffett Field, 94035-0001, USA
[4] CNRS, Université de Nantes, UMR 6554 LETG, Campus du Tertre, Nantes, 44312, France
[5] Laboratoire de Météorologie Dynamique, Ecole Polytechnique, Palaiseau, 91128, France

*Correspondence to*: Kunihiko Kodera (kodera.kk@gmail.com)

**Abstract.** Large changes in tropical circulation, in particular those related to the summer monsoon and cooling of the sea surface in the equatorial eastern Pacific, were noted from the mid-to-late 1990s. The cause of such recent decadal variations in the tropics was studied using a meteorological reanalysis dataset. Cooling of the equatorial southeastern Pacific Ocean occurred in association with enhanced cross-equatorial southerlies, which resulted from a strengthening of the deep ascending branch of the boreal summer Hadley circulation over the continental sector connected to stratospheric circulation. From boreal summer to winter, the anomalous convective activity centre moves southward following the seasonal march to the equatorial Indian Ocean–Maritime Continent region, which strengthens the surface easterlies over the equatorial central Pacific. Accordingly, ocean surface cooling extends over the equatorial central Pacific. We suggest that the fundamental cause of the recent decadal change in the tropical troposphere and the ocean is a poleward shift of convective activity that resulted from a strengthening of extreme deep convection penetrating into the tropical tropopause layer, particularly over the African and Asian continents and adjacent oceans. We conjecture that the increase in extreme deep convection is produced by a combination of land surface warming due to increased $CO_2$ and a reduction of static stability in the tropical tropopause layer due to tropical stratospheric cooling.

## 1 Introduction

Large changes in tropical circulation occurred from the mid-to-late 1990s. These include a slowdown, or hiatus, of global warming in association with a decrease in the tropical eastern Pacific sea surface temperature (SST) (Kosaka and Xie, 2013; England et al., 2014; Trenberth et al., 2014; Watanabe et al., 2014). Changes were also found in the advancement of the onset of the Asian summer monsoon (Kajikawa et al., 2012; Gautam and Regmi, 2013; Xiang and Wang, 2013; Yun et al., 2014) and an increase in precipitation in West Africa over the Sahel (Fontaine et al., 2011; Brandt et al., 2014; Maidment et al., 2015; Diawara et al., 2016). An increase in precipitation in southern Africa was also noted during austral summer (Vizy

and Cook, 2016). In addition to these large-scale circulation changes, variations occurred in mesoscale phenomena such as an increase in Mesoscale Convective Systems (MCSs) over the Sahel (Taylor et al., 2017). Changes in tropical cyclone frequency and intensity over the Arabian Sea (Evan and Camargo, 2011) were also reported. Wang et al. (2012) pointed out that these phenomena are, in fact, related to the above-mentioned early onset of the Asian summer monsoon. A relationship

between tropopause layer cooling and tropical cyclone activity in the Atlantic has also been suggested (Emanuel et al., 2013). Indeed, recent numerical model studies show that cooling of the tropopause impacts the intensity of tropical storms as well as SSTs (Ramsay, 2013; Wang et al., 2014). In this respect, the recent cooling of the tropical tropopause and lower stratosphere from around 2000 (Randel et al., 2006; Randel and Jensen, 2013) should be investigated together with tropical tropospheric change.

The importance of the Pacific Decadal Oscillation (PDO) to decadal changes in global temperature and precipitation has been noted previously (Meehl et al., 2013; Dong and Dai, 2015; Trenberth, 2015). The most recent hiatus in global warming ended around 2013 followed by a strong warming due to an El Niño event in 2015 (Hu and Fedorov, 2017; Liu and Zhou, 2017; Urabe et al., 2017; Xie and Kosaka, 2017). However, the El Niño of 2015/16 differed from the large 1997/98 El Niño,

which involved less warming in the eastern Pacific (Paek et al., 2017), conforming to a trend of increasing intensity of central Pacific-type El Niños (Kao and Yu, 2009; Johnson, 2013). In this sense, the anomalous tropical circulation of the 1990s did not terminate with the hiatus, but persists today. Similarly, the northward shift of the convective zone in boreal summer continues, as shown below.

Multidecadal variations in the atmosphere–ocean coupled mode, such as the Atlantic Multidecadal Oscillation, have also been proposed as a cause of recent changes in tropics (Wang et al., 2013; Kamae et al., 2017). Kamae et al. (2017) studied the impact of SSTs in various oceanic basins on recent trends in monsoon precipitation using a coupled ocean model. They were able to reproduce the recent increasing trend in monsoon rainfall in the Northern Hemisphere (NH), except for the Asian monsoon, with changes in Atlantic SST. Atlantic SSTs, however, have practically no effect on the African or

Australian monsoons in the Southern Hemisphere (SH). Another difference from observations is that the simulated increase in rainfall occurs mainly over the oceans and at low latitudes between the equator and 15°N rather than over continents between around 10°N and 20°N, as observed (see fig. 3 of Kamae et al., 2017). Thus, it is difficult to attribute recent global trends to a regional mode of decadal oceanic variation alone. In this paper, we show that the fundamental cause of the recent decadal trend in the tropics from the mid-to-late 1990s is not the PDO, but rather a strengthening of the deep ascending

branch of the summertime Hadley circulation extending into the stratosphere.

One feature of the recent trend in tropical circulation is a poleward expansion of the tropics (e.g., Davis and Rosenlof, 2012; Lucas et al., 2013; Hu et al., 2018). However, this expansion of the tropics is related to changes in the descending branch of the Hadley cell in the subtropics due to changes in the positions of jet streams and storm tracks (Seidel et al., 2007; Kang and

Polvani, 2011). Thus, studies of tropical expansion focus on phenomena other than a vertical connection in the tropics, which is the focus of this work.

Global climate change involves diverse aspects from the stratosphere to the ocean, from the polar region to the tropics, and from monsoons to severe storms. Each of these elements should be investigated independently in great detail, but their relationships to each other and their roles in global climate change also warrant. Without the latter, we will be unable to see the 'big picture.' Stratospheric variation has generally been treated as a problem separate from recent surface climate change. The goal of this study is to provide a framework for assembling these diverse pieces of the climate-change puzzle by investigating the connection between the atmosphere and ocean in the tropics.

The remainder of this paper is organized as follows. The data used in this study are presented in section 2, and the results of our analysis are given in section 3. A summary and discussion of the causes of recent climate changes in tropics are presented in section 4.

## 2 Data

We use meteorological reanalysis data produced by the Japan Meteorological Agency (JMA), JRA55 (Kobayashi et al., 2015). A large discontinuity was found at the end of the 1990s in a previous reanalysis product, JRA25, when the TIROS Operational Vertical Sounder (TOVS) onboard the National Oceanic and Atmospheric Administration (NOAA) satellite was switched to Advanced TOVS (ATOVS; Li et al., 2000). This discontinuity has largely been removed in the JRA55 reanalysis (Kobayashi et al., 2015).

Outgoing longwave radiation (OLR) data provided by NOAA are widely used in analyses of convective activity in the tropics. In the present study, we use monthly mean OLR data (1° × 1° latitude–longitude resolution) derived from the High-Resolution Infrared Radiation Sounder (HIRS) (Lee et al., 2007), available at ftp://eclipse.ncdc.noaa.gov/cdr/hirs-olr/monthly/. An analysis of the precipitation is performed using Global Precipitation Climatology Project (GPCP) monthly mean data version 2.3 (Adler et al., 2003). Monthly mean gridded SST data from COBE with 1° × 1° grid cells compiled by the JMA (Ishii et al., 2005) are used for the study of ocean surface change.

The climatology is defined here as the 30-year mean from 1981 to 2010. Seven El Niño events after 1979 are identified by the JMA based on 6-monthly mean SSTs in the Niño 3 sector (5°S–5°N, 150°W–90°W; available at http://ds.data.jma.go.jp/gmd/tcc/tcc/products/elnino/ensoevents.html). In this study, we define the NH cold seasons of 1982/83, 1986/87, 1991/92, 1997/98, 2002/03, 2009/10, and 2015/16 as El Niño winters.

Tropical overshooting clouds (COV) were identified using the diagnostics developed by Hong et al. (2005), which are based on brightness temperature differences measured by three high-frequency channels of the Advanced Microwave Sensing Unit (AMSU) module B or the Microwave Humidity Sensor (MHS). Data are from NOAA and MetOp satellites with periods of 2007–2013 for MetOp-A and 2014–2017 for MetOp-B. Their equatorial crossing times are nearly identical (see fig. 1 of Funatsu et al., 2016). The original data calculated on a 0.25° × 0.25° grid were resampled to a coarse one of 2.5° × 2.5° grid for plotting. The number density of COV is defined as the total number of COV detected in each 2.5° × 2.5° bin divided by the MetOp–MHS total pixel number to remove sampling bias, with units of parts-per-thousand.

The maximum cloud-top height (CTH) of convective clouds is calculated every 3 hours on a 0.25° × 0.25° grid based on brightness infra-red (IR) data from geostationary satellites. Convective clouds are identified using Tropical Rainfall Measuring Mission (TRMM) data, and CTHs were calibrated by CLOUDSAT and CALIPSO measurements (Pfister et al., 2018). Occurrence frequency of CTH > 17km for each 2.5° × 2.5° grid is calculated as (number of 3 hourly cloud top data satisfying given condition CTH >17km, in that grid box) / (total number of 3 hourly cloud top data). Occurrence frequency is expressed by %.

## 3 Results

### 3.1 Deep branch of the Hadley circulation

The Hadley circulation, the most fundamental tropical circulation, is usually characterized by its intensity and width or latitudinal extent in the mid-troposphere (e.g., Nguyen et al., 2013). However, the Hadley circulation, or the mean meridional circulation driven by convection, extends to higher levels in summer. Here we first examine climatological aspects of the residual mean-meridional circulation during July–September diagnosed according to the method of Iwasaki (Iwasaki, 1992; Kobayashi and Iwasaki, 2016) (Fig. 1a). Stream lines are plotted approximately with logarithmic scaling. The region of major upwelling in the lower troposphere is located over the inter-tropical convergence zone (ITCZ) around 5°N–10°N. However, its associated ascending branch (blue stream lines in Fig. 1a) is shallow, and the air diverges near the top of the troposphere and descends in the subtropics. In contrast, the ascending branch around 15°N extends into the tropical tropopause layer (TTL) and is connected to the upwelling branch of the stratospheric Brewer–Dobson (BD) circulation (red stream lines in Fig 1a).

Because the upwelling of the deep branch is driven by convective activity, we consider that this constitutes a part of the Hadley circulation, although the Hadley circulation is usually treated as a cell confined within the troposphere. The horizontal distributions of the shallow and deep ascending branches are characterised by the horizontal air divergence field and convective activity (or precipitation) displayed in the left- and right-hand panels of Fig. 1, respectively. The shallow branch is characterised by a convergence zone near the surface at 925 hPa and its mirror image in the divergence field in the

upper troposphere at 250 hPa. This feature is particularly clear over the oceanic sector where a band of heavy precipitation occurs along the ITCZ. In contrast, a near-surface convergence zone is not clearly defined over the continental sector, and heavy precipitation tends to occur over coastal regions on the west side of the continents. The discrepancy between the convergence zone and the rain band is particularly large over the African continent (Nicholson, 2018). The deep ascending branch is associated with deep penetrating convection, which occurs over regions of high moist static energy around the continental monsoon sector.

## 3.2 Surface variation vs. OLR

The recent poleward shift of the convergence zone during boreal summer is identified from the July–September mean anomalous OLR during 1999–2016 from the 30-year climatology (1981–2010; Fig. 2a). Increases in convective activity occur over summer monsoon regions, Africa, Asia, and Central America. It should be noted that these continental convectively active regions coincide with a climatological deep ascending branch, as shown in Fig. 1. Because the deep ascending branch in boreal summer is located primarily over continents north of the equator, its enhancement manifests as a poleward shift in convective activity. This feature is more easily seen in the zonal-mean OLR field shown in Fig. 2b. Climatological OLR peaks around 10°N, whereas the anomalous OLR of the recent period has a maximum around 15°N.

There is a close relationship between the location of the seasonally varying tropical convergence zone and cold tongues in the tropical oceans. Convective activity shifts northward during boreal summer. Accordingly, cross-equatorial winds west of the American and African continents increase, which leads to a decrease in SSTs along coastal regions during boreal summer as a part of a seasonal cycle. Thus, the primary cause of cold tongues in tropical SSTs is the shape of the continents, the air–sea interaction, and the location of the rising branch of the Hadley circulation, as described by Xie and Philander (1994) and Xie (2004). Therefore, changes in the zonal-mean meridional circulation, such as those shown in Fig. 2, can affect equatorial eastern Pacific SSTs.

To investigate whether the northward shift in the convective zone is driven by the PDO, anomalous OLR during the two periods of neutral and negative phases of the PDO is shown in Fig. 2c and d with anomalous SSTs during those periods (Fig. 2e and f). A characteristic horseshoe pattern in northern Pacific SST is evident during the negative phase of the PDO. Anomalous OLR indicates that convective activity is enhanced along 15°N–20°N irrespective of the phase of the PDO, except for the sector under the direct influence of the PDO in the eastern Pacific, where cooling is greater during the negative phase. However, even during the neutral phase of the PDO, negative anomalies in SST exist in the tropics west of South America. This suggests that SST cooling west of South America is not driven solely by the PDO, but is related to the strengthening and northward shift of convective activity.

The impact of the recent decadal variation in convective activity on SST is depicted in Fig. 3. The spatial structure of the recent decadal trend varies with the season. The top panels show the 1999–2016 mean anomalous OLR during (a) JAS and (b) OND. Because the response of SST follows the atmospheric circulation, anomalous SSTs during the following month (i.e., August, September and October (ASO) and November, December and January (NDJ)) are displayed in Fig. 3c and d, respectively. During JAS, the anomalous cross-equatorial flow west of South America intensifies because of a poleward shift in convectively active regions. The cross-equatorial flow changes from westward to eastward when it crosses the equator, following the change in sign of the Coriolis force. This results in a strengthening of the climatological easterlies in the SH and enhances anomalous convergence near New Guinea. In contrast, easterlies are weakened in the NH, which explains the warming (cooling) north (south) of the equator. Such a meridional seesaw of anomalous SSTs and cross-equatorial flow suggests an important role for wind−evaporation−SST (WES) feedback (Xie and Philander, 1994) in recent trends. The centre of anomalous negative OLR moves to the equatorial eastern Indian Ocean from boreal summer to autumn, which results in a strengthening of anomalous easterlies over the equatorial central Pacific and a westward extension of low SSTs over the equator.

In the analysis above, two different features of decadal variability are evident, over continental and oceanic sectors, which correspond to the locations of the deep and shallow ascending branches of the Hadley circulation, respectively. Here we examine variations over the African continental (10°W–40°E) and Pacific Ocean (170°W–120°W) sectors to reveal the most prominent characteristics in each region. The climatological annual cycle in zonal-mean pressure vertical velocity at 300 hPa is depicted in Figs 4a and 5a. A region of enhanced convective activity migrates north and south over the African continent following the seasonal variation of solar heating. It should be noted that the evolution of the convective zone includes a jump during the summer monsoon season (Hagos and Cook, 2007). Over the Pacific Ocean, the convective zone shows only a small latitudinal displacement and is located in the NH near 5°N–10°N throughout the year. Latitude–time cross-sections of the 3-monthly mean anomalous (departures from climatology) 300-hPa vertical velocities are shown for February 1979 to November 2016 over the African sector in Fig. 4b. The vertical velocity increases from the mid-1990s in both hemispheres around 10°−20° in latitude, corresponding *to the deep ascending branch* of the summertime Hadley circulation. Accordingly, the annual mean precipitation over Africa has increased during the recent period (1999−2016) in both hemispheres over the Sahel and Namibia (Fig. 4c).

Over the Pacific Ocean sector (Fig. 5b), strong upward motion appears over the equator when El Niño events occur. This has been identified as an effect of the ENSO on the ITCZ (Waliser and Gautier,1993). The anomalous region of upward motion, however, tends to remain north of the equator after 1999. Accordingly, the annual mean anomalous precipitation during the recent period shows a large increase near 5°N−10°N, the mean position of the ITCZ over the ocean, but decreases over the equator and the SH (Fig. 5c). This manifests as a narrowing and intensification of the ITCZ in recent decades, with little change in latitudinal position (Lucas et al., 2013; Wodzicki and Rapp, 2016). The change over the ocean sector is related to

the varying strength of the cross-equatorial winds (Fig. 5d). After 1999, although SSTs increased over the equator during El Niño events, anomalous northward winds remained strong and convective activity tended to remain in the NH.

Latitude−time sections of 3-monthly anomalous SSTs in the Niño 3.4 sector (Fig. 5g) indicate little change in latitudinal structure. Figure 5e shows a longitude–time section of the anomalous OLR over the equatorial SH (0°−10°S). The effects of cooling of the eastern equatorial Pacific in the SH can also be seen in structural changes in El Niño/Southern Oscillation (ENSO) phenomena after 1999. Convective activity greatly increases over the Pacific during El Niño events before 1999. However, after 1999, Pacific convective activity is suppressed and an increase in convective activity during El Niño is apparent only over the central Pacific. In contrast, convective activity west of 160°E over the Maritime Continent generally increases after 1999. Such changes are likely related to a decadal change in anomalous zonal winds over the tropical SH (10°S−5°N; Fig. 5f), which in turn is connected to increased cross-equatorial southerlies through Coriolis-force effects.

### 3.3 Stratosphere–Troposphere coupling

In addition to changes in the troposphere, Abalos et al. (2015) identified an increasing trend in the tropical lower stratospheric upwelling at 70 hPa, together with an increasing number of sudden stratospheric warming events. Because the upwelling in the deep ascending branch of the Hadley circulation extends to the stratosphere (Fig. 1a), it is expected that an interaction between tropospheric and stratospheric circulation may exist near the TTL.

To investigate this possibility, a singular value decomposition (SVD) analysis (Kuroda, 1998) was conducted using the normalized covariance matrix between the zonal-mean pressure vertical velocity and the horizontal distribution of the OLR (30°S–30°N) during boreal summer (JAS) (Fig. 6). The value at each grid point was weighted by the vertical-layer thickness and the cosine of the latitude in the meridional direction. To investigate the relative importance of zonal-mean vertical velocity at various altitudes, the SVD calculations were performed with zonal-mean pressure vertical velocities at the (a) 150–50 hPa and (b) 1000–300 hPa levels. To obtain a general view for the entire troposphere and stratosphere, the heterogeneous correlation of the vertical velocity was extended to a height range of 1000 to 10 hPa. Variations related to a decadal variation and the ENSO cycle appear as the first two modes. Here we show only variations related to the decadal change. Increasing trends in coefficients are evident in both cases; however, the zonal mean pressure vertical velocity ($\omega$) of 15°N–20°N is more closely related to variations around the TTL (Fig. 6a) than to variations in the lower troposphere (Fig. 6b). Accordingly, variations in OLR around 15°N over the African–Asian sector are more closely related to vertical velocity in the TTL. In the SVD of tropospheric vertical velocity, the heterogeneous correlation with OLR indicates a relationship with convective activity over the equatorial Pacific north of the equator, which is confined mainly to the troposphere. These results suggest a stronger connection between variations in convective activity around the deep ascending branch of the Hadley circulation and lower stratospheric circulations. This is consistent with results from previous work (Eguchi et al., 2015; Kodera et al., 2015).

### 3.4 Variations over continents and oceans

Tropospheric zonal-mean vertical velocity has a relatively small connection with the horizontal distribution of OLR. This may result from the fact that regional-scale variations dominate in the lower troposphere because of surface geography. Therefore, meridional sections of standardized mean JAS 1999–2016 anomalous pressure vertical velocity were calculated for several sectors (Fig. 7). If a normal distribution is assumed, absolute values of 17-year mean standardized anomalies (Fig. 8) that are larger than 0.5 are statistically different from 0 at the 95% confidence level.

The top panel in Fig. 8 shows the zonal-mean field, which is comparable to that extracted by SVD analysis shown in Fig. 6a. Contours indicate the climatology. The middle panels are the same as the top panel, but divided into two parts: (left) an African–Asian continental sector (30°W–130°E) and (right) a Pacific–Atlantic oceanic sector (130°E–330°E). A strengthening of upward velocity in the TTL and lower stratosphere occurs in the continental sector, adding to the northward shift in the troposphere, whereas in the oceanic sector a strengthening in vertical velocity occurs around 5°N–10°N without a latitudinal shift. If we limit the continental sector to the African continent (20°W–20°E) to exclude the influence of the Indian Ocean, the above-mentioned continental characteristics become even clearer (Fig. 8, bottom-left). Over the oceanic sector, an increase in vertical velocity occurs around 5°N (Fig. 8, bottom-right), but in the western Pacific sector (130°E–170°E) the upward velocity develops primarily south of the equator (10°S–0°; Fig. 8, bottom-centre). We also note that the climatological vertical velocity in the western Pacific sector is essentially confined to the lower troposphere over the equatorial SH (10°S–0°). This observation can be attributed to the fact that convergence occurs over the warm ocean east of New Guinea (Fig. 3c). This result indicates that despite a variety of profiles among the sectors, the zonal-mean vertical field in the TTL primarily follows variations over the African–Asian continental sector.

Continuity in a zonally averaged field does not necessarily mean actual continuity at each location, as is evident from the above analysis. To investigate continuity within the rising branch of the Hadley circulation from the upper troposphere to the stratosphere in more detail, longitude−height sections of the normalized anomalous pressure vertical velocities averaged over latitudes of 10°−20° in the summer hemisphere are displayed in the top panels of Fig. 8a and b. The bottom panels show the distributions of climatological (2007−2017) COV occurrence frequency in the same latitudinal zone. An increasing trend in upwelling occurs over the continental sector, particularly where COVs are frequent. These characteristics are commonly seen in both summer hemispheres. The contrast between the continental and oceanic sectors is clearer in the SH where the distribution of land surfaces is simpler. Because COV occurs in deep convective clouds penetrating into the TTL beyond the level of neutral buoyancy, such increased vertical velocity in the TTL over the region of frequent COV seems reasonable. It should also be noted that a connection between COV and vertical velocity in the tropical lower stratosphere on a daily scale has been identified in a previous study by Kodera et al. (2015).

To investigate stratosphere-related variations in the troposphere, the JAS mean pressure vertical velocity ($\omega$) at 30 hPa averaged over the tropical SH (0°–25°S) is chosen as the index of stratospheric mean meridional circulation ($I_\omega$; Fig. 9a). The correlation coefficient between $I_\omega$ and zonal-mean $\omega$ at each grid point (Fig. 9b) shows a correlation pattern quite similar to the SVD mode in Fig. 6a. To reveal the relationship between the interannual variation and climatology, the stream function from Fig. 1a is displayed as contours in Fig. 9b and c. It is clearly seen that the variation in stratospheric upwelling (Brewer–Dobson circulation) is well connected to the upwelling of the deep ascending branch of the Hadley circulation, similar to that in climatology.

The correlation between $I_\omega$ and zonal-mean temperature at each grid point from 90°S to 90°N is shown in Fig. 9c. Tropical upwelling is not only related to cooling in the tropics and the summer hemisphere, but to warming in the downwelling region around the winter polar stratosphere. This suggests the dynamic nature of recent tropical stratospheric cooling. Stratospheric upwelling is also connected with convective activity along 15°N−20°N (Fig. 9d), as discussed above. Correlation coefficients between $I_\omega$ and 925-hPa zonal and meridional winds at each grid point are shown as arrows in Fig. 9e. An increase in cross-equatorial winds in the eastern Pacific and Atlantic is observed. The impact of near-surface wind variations on SST can be seen in the lagged correlation with SST in Fig. 9e. Cooling in the equatorial eastern Pacific is largest with a time lag of 5 months (i.e., during December, January, and February, DJF), consistent with the development of La Niña-like SSTs during boreal autumn (Fig. 3).

Results of the above analyses are summarized schematically in Fig. 10 (left). Accordingly, we selected four key variables that can be considered fundamental to the recent tropical trends: (a) tropical lower-stratospheric temperatures in early summer (temperature at 70 hPa averaged over 20°S–20°N from 16 July to 16 August); (b) pressure vertical velocity at the bottom of the TTL (150 hPa) in August; (c) August–October mean southward winds south of the equator (10°S–0°) in the western hemisphere (180°W–0°); and (d) time tendency of SST from early summer (May–July) to late autumn (October–December) in the tropical Pacific west of the South American continent (15°S–5°S, 100°W–80°W). Time series of these four variables (a–d) are displayed in Fig 10 (right). When all four variables are negative (red dots), we define this as a negative event. Similarly, when all variables are positive (black dots), it is defined as a positive event. All six positive events occurred within the first 14 years, whereas all seven negative events occurred during the last 13 years. A chi-squared test was conducted to examine whether such distributions of events can occur by chance, by dividing the whole 39 years into three equal 13-year periods. The result ($\chi^2 = 23$) indicates that the probability of such distributions occurring by chance is less than 0.1%. Therefore, there is a statistically significant trend towards negative events in recent decades.

However, the key question here is whether there is a causal relationship among the variables. We introduced a seasonal variation in the selection of the variable from the period of stratospheric cooling at the end of July and to the cooling of the ocean from summer to autumn. This time evolution tentatively suggests a causality among the variables. To further

investigate the coupling process between the stratosphere and the troposphere, a case study is presented in the next subsection.

## 3.5 Stepwise transition

Decadal changes do not necessarily comprise only decadal time-scale variations, but can also include seasonality; i.e., seasonally phase-locked decadal variations. Thus, decadal trends can include sharp stepwise seasonal transitions, similar to the monsoon onset. Summer 2010 is typical of summers in recent decades (Fig. 10); however, a sharp transition from a spring state to a summer state occurred during this year. The evolution of anomalous monthly mean tropical (20°S–20°N) temperature in the lower stratosphere and upper troposphere is shown in Fig. 11a for March to October 2010. Anomalous negative temperatures associated with easterly winds of the quasi-biennial oscillation (QBO) gradually descend in the stratosphere during this period. However, a sudden decrease in temperature occurred in July and lasted until September. Simultaneously, anomalous negative OLR that in spring was located near 5°N shifted to 15°N in July. The horizontal distributions of seasonal mean anomalous OLR for April–June and July–September are shown in Fig. 11c and d, respectively, along with the zonal-mean field in the right-hand panels. Anomalous OLR in summer 2010 (Fig. 11d) has a similar pattern to the recent 17-year mean summer anomalies (Fig. 2a) with anomalous negative OLR along 15°N over the African–Asian sector.

This anomalous OLR pattern in summer 2010 abruptly formed in mid-July and persisted throughout the summer. Figure 12a and b shows a time series of eddy heat flux at 100 hPa in the extratropical SH (45°S–75°S), and a height–time section of the tropical temperature tendency from 25 June to 29 July 2010. The rapid decrease in stratospheric temperature in mid-July was induced by enhanced planetary wave activity in the winter hemisphere. Height–latitude sections of 7-day mean standardized pressure vertical velocity (Fig. 12c) show that stratospheric upwelling increased around 14 July when the temperature tendency changed from positive to negative. To investigate deep convective activity reaching the TTL, 7-day mean occurrence frequencies of convection with CTH higher than 17 km and between 15 and 17 km are shown in Fig. 12c as solid lines, above and below the 100-hPa level, respectively. When the temperature tendency changed from positive to negative in the stratosphere around 14 July, extreme deep convective clouds with CTH > 17 km greatly increased around 15°N. In the following period (around 21 July), peak occurrence frequency slightly decreased, but extreme deep clouds spread over a wider latitudinal range, from 10°N to 25°N. Variations in the amplitude of less-deep clouds (15 km < CTH ≤ 17 km) are less prominent and changes appear primarily in their latitudinal distributions. It should be noted that variations in the distributions of deep convective clouds around the tropopause level fits well to the evolution of anomalous pressure vertical velocities derived by reanalysis. Anomalous upwelling is enhanced in the upper troposphere from 14 July and extends to the lower troposphere on 21 July. Anomalous upwelling also appears as enhanced anomalous negative OLR. A rapid increase in convective activity around 15°N can be clearly seen in the horizontal distribution of the 7-day mean anomalous OLR field, in

particular over the African–Asian sector (Fig. 13). This pattern persisted throughout the rest of the summer, thus creating a similar pattern in the seasonal mean field, as evident in Fig. 11d.

## 4. Summary and discussion

Characteristics of recent tropical circulation changes from the middle to the end of the 1990s can be summarized as follows. Decadal cooling in the eastern Pacific is related to an increase in cross-equatorial winds, and easterlies in the tropical SH, which are themselves related to a strengthening of convective activity (Fig. 3) around the climatological deep ascending branch of the Hadley circulation during boreal summer over the African–Asian sector (Fig. 1). In addition, a correlation analysis (Fig. 9) indicates that these variations in convective activity and SST are related to vertical velocity near the tropopause.

It is difficult to demonstrate statistically a causal relationship among variables having large trends, such as (a) lower stratospheric temperature, (b) upwelling in the TTL, (c) cross-equatorial near-surface winds, and (d) time tendency of SST from boreal summer to autumn. Nevertheless, time lags introduced in selection of variables from summer to autumn demonstrate that the above processes are related, as shown schematically in the left panel of Fig 10.

The connection between tropical stratospheric cooling and extreme deep convection in the deep ascending branch of the Hadley cell occurred in mid-July 2010. Because this anomalous pattern persisted throughout the summer, this process is likely responsible for the seasonally locked decadal changes in stratosphere–troposphere coupling. Changes in convective activity in the deep ascending branch of the Hadley cell can modulate cross-equatorial winds. The relationship between convective activity and cooling of the tropical eastern Pacific can be explained through changes in cross-equatorial winds involved in a wind−evaporation−SST (WES) feedback (Xie, 2004). Accordingly, a combination of these two processes can be used as a working hypothesis for recent tropical changes, as shown in Fig. 10.

Although their period of observation may be too short (10 years of Atmospheric Infrared Sounder data), Aumann and Ruzmaikin (2013) reported that tropical deep convection over land shows an increasing trend, whereas that over oceans shows a decreasing trend. Furthermore, Taylor et al. (2017) showed that intense mesoscale convective systems in which cloud-top temperatures were lower than −70°C largely increased over the Sahel since 1982. A temperature of −70°C corresponds to the air temperature at ~150 hPa. This means that extreme deep convection penetrating into the TTL largely increased over the African continent, consistent with the present study.

A poleward shift in the convective zone occurred because of enhanced convective activity in the deep ascending branch of the Hadley circulation over the continental sector. This phenomenon studied here is different from the shift of the ITCZ over the oceanic sector due to perturbations in the atmospheric energy balance such as discussed by Schneider et al. (2017).

The increasing trend in Earth's surface temperature is generally attributed to an increase in greenhouse gases, such as $CO_2$ (IPCC, 2013). Such a change in radiative forcing may explain the global characteristics of recent changes. The effect of increased $CO_2$ can be divided into a direct radiative effect and an indirect effect through changes in SST. Model experiments have shown that the direct radiative effect of $CO_2$ increases tropical upward motion, particularly over the Sahelian sector, whereas it suppresses upwelling over the oceanic sector in the Pacific (see fig. 8 of Gaetani et al., 2016). An increase in $CO_2$

raises the Earth's surface temperature, but decreases stratospheric temperatures. Note, however, that recent cooling in the lower stratosphere–tropopause region is also due to a dynamic effect (Abalos et al., 2015). Further investigation is needed to determine whether the stratosphere is merely passively responding or playing an active role in tropospheric circulation change. Here we emphasized that stratospheric change should be considered together with tropospheric change.

**5 Data availability**

Datasets used in this paper are all publicly available. Meteorological reanalysis datasets created by JMA (JRA55) are available from http://search.diasjp.net/en/dataset/JRA55. The COBE monthly mean SST dataset can be obtained from the JMA website (http://ds.data.jma.go.jp/tcc/tcc/products/elnino/cobesst/cobe-sst.html). Monthly mean HIRS OLR data can be obtained from NOAA by FTP (ftp://eclipse.ncdc.noaa.gov/cdr/hirs-olr/monthly/). The GPCP monthly mean precipitation dataset can be obtained from the NOAA website (https://www.esrl.noaa.gov/psd/data/gridded/data.gpcp). The AMSU/MHS

data are available at NOAA's Comprehensive Large Array Data Stewardship System. In this work, AMSU/MHS raw data were obtained with support from the INSU-CNES French Mixed Service Unit ICARE/climserv/AERIS and accessed with the help of ESPRI/IPSL. The cloud-top height dataset is available through L. Pfister (leonhard.pfister@nasa.gov).

**Acknowledgements**

This work was supported in part by Grants-in-Aid for Scientific Research (25340010, 26281016, and 16H01184) from the

Japan Society for the Promotion of Science. Preliminary analysis of this study were made using Interactive Tool for Analysis of Climate System, ITACS, provided by the Japan Meteorological Agency.

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

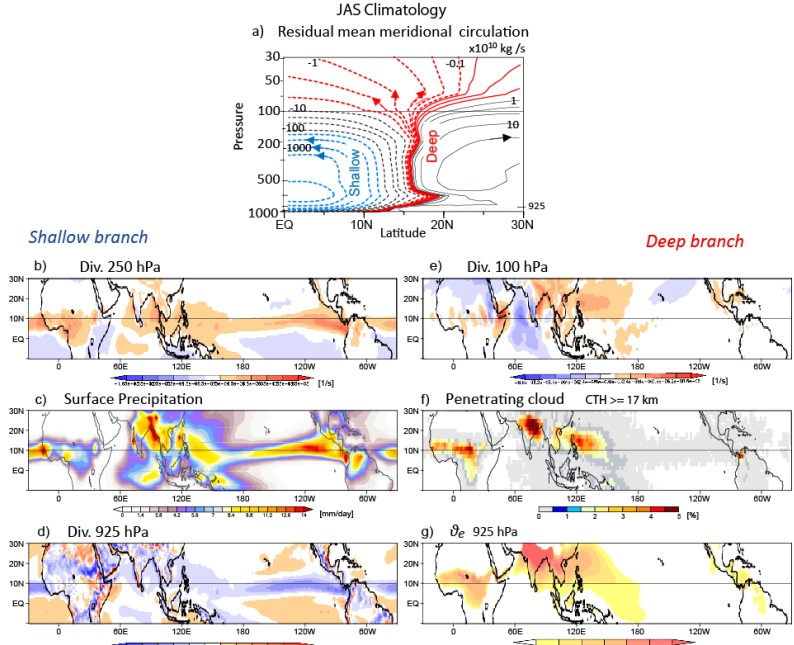

**Figure 1:** July–September climatology: (a) mass stream function of mean meridional residual circulation (contours: ± 0.1, 0.2, 0.5, 1, 2, 5, 10, and 20 × 10$^{10}$ kg s$^{-1}$; red and blue stream lines indicate the deep and shallow ascending branches, respectively), (b) horizontal divergence of air at 250 hPa, (c) surface precipitation from GPCP, (d) divergence at 925 hPa, (e) divergence at 100 hPa, (f) occurrence frequency of deep convection with CTH ≥ 17 km, and (g) equivalent potential temperature at 925 hPa. Climatology is for 1981–2010 except for (e), which is for 2005–2017.

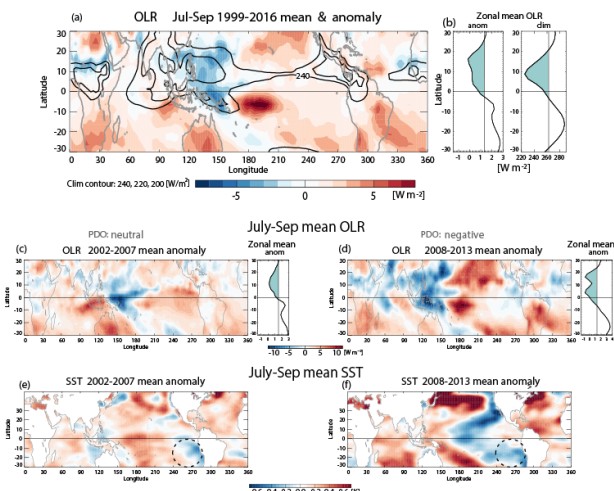

**Figure 2:** (a) Climatological (1981–2010) July–September mean OLR (contours: 240, 220, and 200 W m$^{-2}$) and anomalous July–September OLR (departures from climatology) during 1999–2016 (colour shading); (b) zonal-mean profiles of (a): anomalies from climatology (left) and climatology (right); (c) anomalous OLR as in (a) and (b, left) but for 2002–2007; (d) anomalous OLR as in (a) and (b, left) but for 2008–2013; and anomalous July–September SST (departures from climatology) for (e) 2002–2007 and (f) 2008–2013.

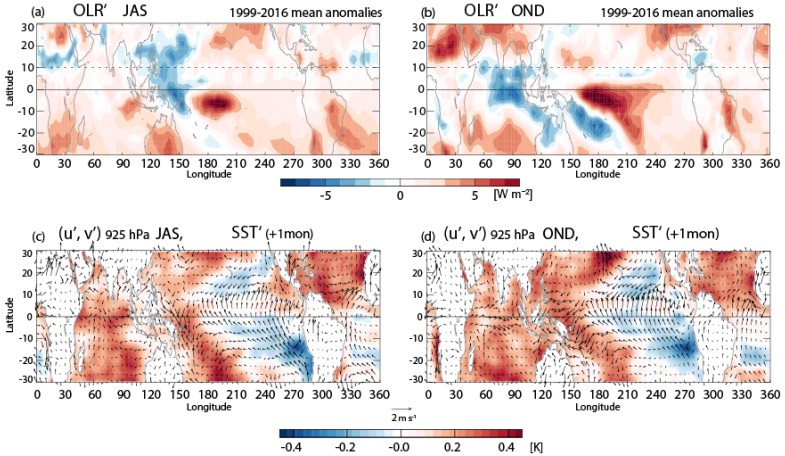

**Figure 3: (a) JAS and (b) OND mean anomalous OLR for 1999–2016; (c) JAS and (d) OND mean anomalous horizontal winds at 925 hPa (arrows) for 1999–2016 superimposed on anomalous SSTs (colour shading) with a one-month lag (i.e., ASO and NDJ, respectively).**

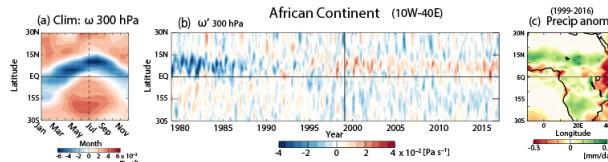

**Figure 4: (a) Latitude–time section of the climatological zonal-mean pressure vertical velocity at 300 hPa averaged over the African sector (10°W–40°E); (b) latitude–time section of monthly mean anomalous pressure vertical velocity from February 1979 to November 2016; and (c) latitude–longitude map of annual mean anomalous precipitation during 1999–2016 over the African sector. A three-month running mean is applied in (b).**

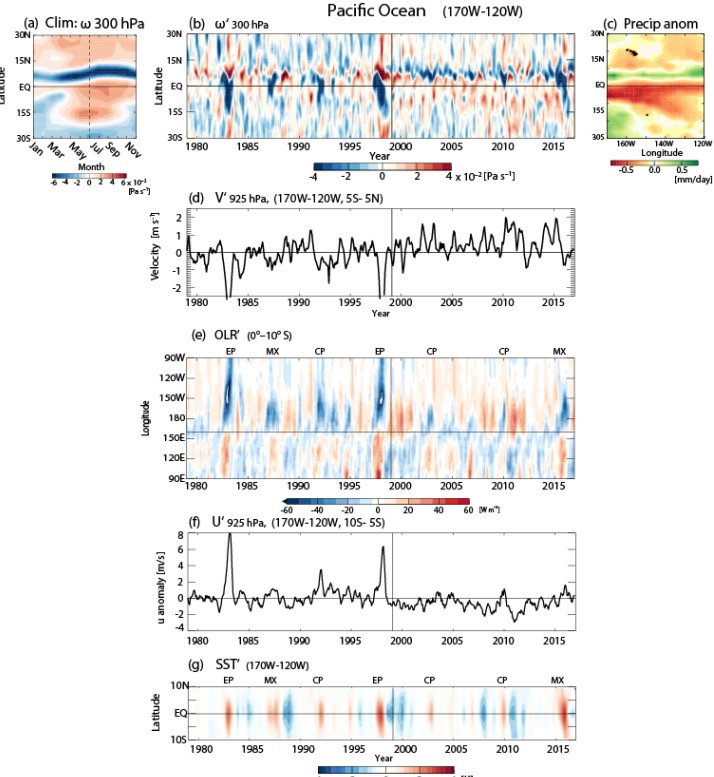

**Figure 5: (a–c) Same as Fig. 4 but for the eastern Pacific Niño-3.4 (170°W–120°W) sector; (d) monthly mean anomalous meridional wind component around the Equator (5°S–5°N) over the Niño 3.4 sector; (e) similar to (a), but for the time–longitude section of OLR around the Equator (5°S–5°N) over the Indian Ocean–Pacific sector; (f) same as (d), but for the zonal wind component in the tropical SH (10°S–5°S); and (g) monthly mean anomalous SST over the Nino 3.4 sector. Eastern Pacific (EP), central Pacific (CP), and mixed-type (MX) El Niño events are indicated (Peak et al., 2017).**

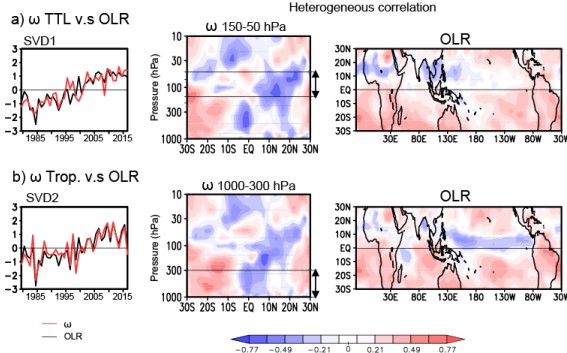

**Figure 6: Singular value decomposition analysis of the zonal-mean anomalous pressure vertical velocity and anomalous OLR (0°–360°E) in the tropics (30°S–30°N) during JAS from 1979 to 2016: (a) SVD 1 of pressure vertical velocity around the TTL (50–150 hPa) and (b) SVD 2 of pressure vertical velocity in the troposphere (1000–300 hPa). From left to right, panels show time coefficients, heterogeneous correlation maps of pressure vertical velocity over 1000–10 hPa, and heterogeneous correlation maps of OLR. Arrows indicate the levels used in the SVD calculations.**

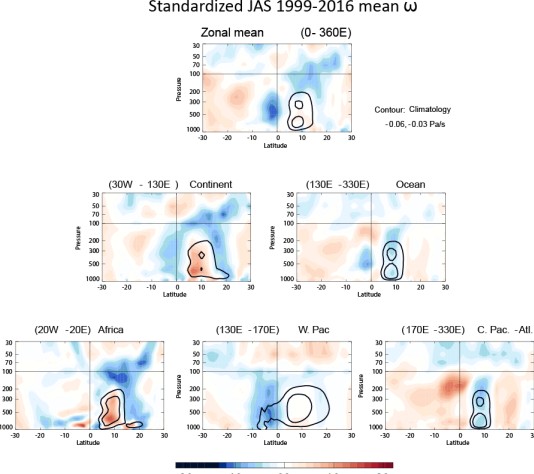

**Figure 7: (top)** Standardized anomalous OLR in JAS 1999–2016 (departures from 1981–2010 climatology)—the climatological JAS mean, zonal-mean pressure vertical velocity is indicated by contours (–0.06 and –0.03 Pa s⁻¹); **(middle)** same as the top panel, but for (left) the African–Asian continental sector (30°W–130°E) and (right) the Pacific–Atlantic oceanic sector (130°E–330°E); and **(bottom)** same as the top panel, but for (left) the African continental sector (20°W–20°E), (centre) the Western Pacific sector (130°E–170°E), and (right) the Central Pacific–Atlantic sector (170°E–330°E).

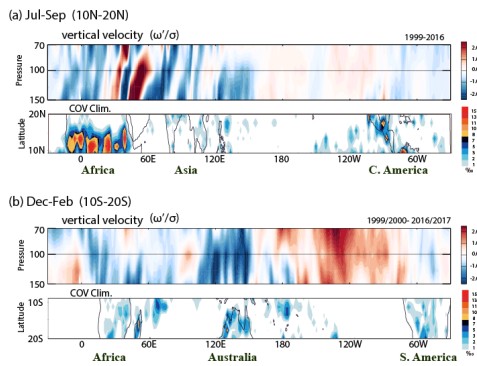

**Figure 8: (a) (top)** Height–longitude section of the standardized (with respect to the interannual variation) anomalous pressure vertical velocity averaged over 10°N–20°N during boreal summer (JAS) 1999–2016; **(bottom)** climatological (2007–2017) occurrence frequency of convective overshooting (COV) in the same latitudinal zone (units of parts per thousand); and **(b)** as in (a) but for 10°S–20°S during austral summer (DJF).

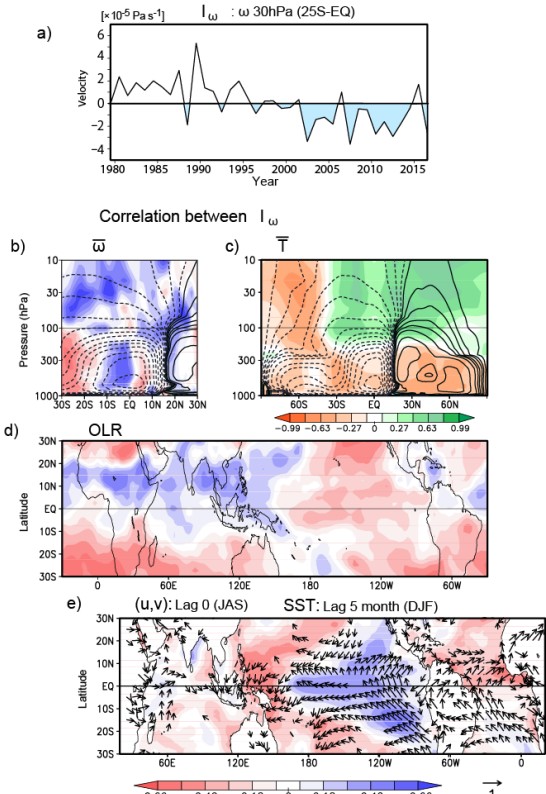

**Figure 9: (a)** Time series of JAS mean pressure vertical velocity ($\omega$) at 30 hPa averaged over 0°–25°S as an index for tropical stratospheric vertical velocity ($I_\omega$); correlation coefficients between $I_\omega$ and **(b)** zonal-mean $\omega$, **(c)** zonal-mean temperature $T$ at each grid, **(d)** OLR, and **(e)** horizontal winds at 925 hPa (arrows). A lagged correlation with DJF mean SST is also presented by colour shading in **(e)**. Contours in **(b)** and **(c)** indicate the climatological residual mean meridional circulation in JAS. Solid and dashed lines indicate clockwise and counter clockwise directions, respectively.

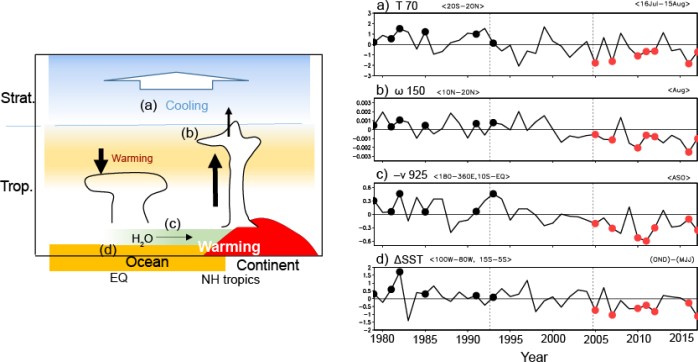

**Figure 10: (left)** Schematic of recent changes in the tropics (see text), in which the labels **(a)** to **(c)** indicate the location of the variable shown in the right panels; **(right)** time series of four key variables as departures from the climatology: **(a)** lower stratospheric temperature, **(b)** upwelling in the TTL, **(c)** cross-equatorial near-surface winds, and **(d)** time tendency of SST from summer to autumn. Black and red dots indicate years when the four variables are of the same polarity (positive and negative, respectively).

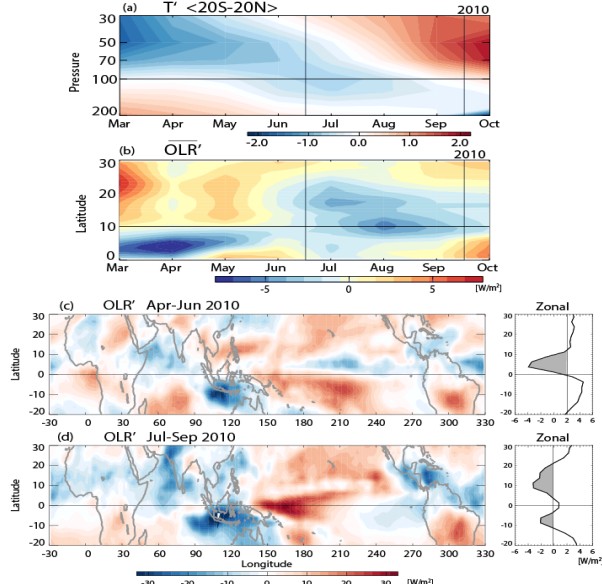

**Figure 11: (a) Height–time section of monthly mean anomalous tropical (20°S–20°N) temperature from March to October 2010; and (b) same as (a) but for a latitude–time section of zonal-mean anomalous OLR. Horizontal distributions of 3-monthly mean anomalous OLR for (c) April–June and (d) July–September 2010 with zonal-mean fields are shown in the right-hand panels.**

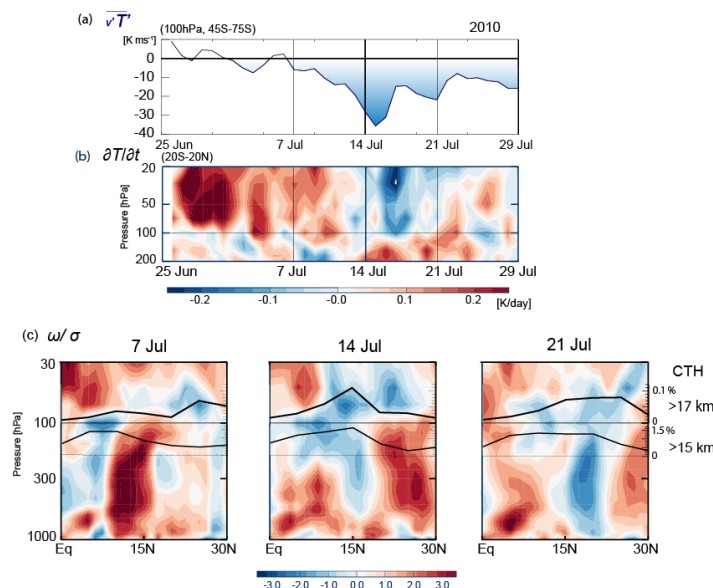

**Figure 12: (a) Daily mean northward eddy heat flux at 100 hPa averaged over 45°S–75°S from 26 June to 29 July 2010; (b) same as (a) but for a height–time section of tropical (20°S–20°N) temperature tendency in the upper troposphere–lower stratosphere; (c) same as (b) but for 7-day mean standardized pressure vertical velocity, for (from left to right) 7, 14, and 21 July—solid lines in**
10  **each panel show the latitudinal distribution of the 7-day mean fractional occurrence frequency (%) of deep convection with CTH > 17 km (upper) and 15 km < CTH ≤ 17 km. The dates indicate the central date of the 7-day mean.**

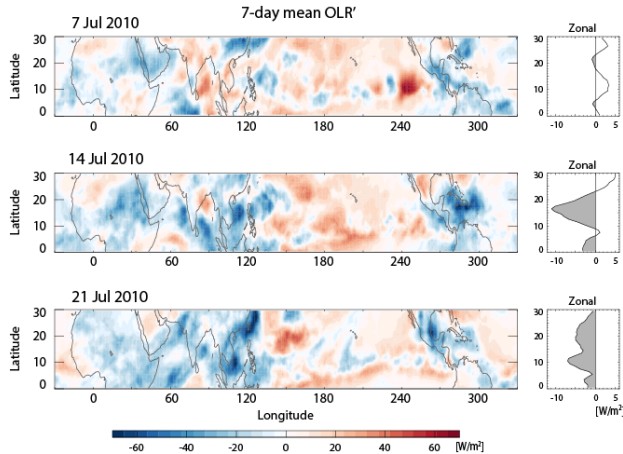

**Figure 13: Same as Fig. 11c but for the 7-day mean anomalous OLR for 7, 14, and 21 July 2010, from top to bottom, with zonal-mean fields in the right-hand panels. The dates indicate the central date of the 7-day mean.**