# Peer review of "Implication of tropical lower stratospheric cooling in recent trends in tropical circulation and deep convective activity"

_Atmospheric Chemistry and Physics, 2018_

## Referee Comment (RC1) · Anonymous Referee #1 · 11 Mar 2018

The global warming hiatus has drawn attention to climate anomalies during the early 2000s. The authors examined OLR anomalies during 1999-2016 and identified a decrease in zonal mean OLR near the climatological ITCZ north of the equator during the period. Their SVD analysis showed that it is part of a global tropical mode with a principle component dominated by a trend during the satellite era after 1979. They further discussed the changes in vertical velocity, near-surface wind and SST. The coherent change patterns the authors identified—the intensified ITCZ and SVD mode with a pronounced trend—are quite interesting and potentially important.

The paper may eventually be publishable in ACP but the current manuscript suffers

from poor organization interrupted by unsubstantiated claims on a wide range of issues (some examples below), some marginally related to the central theme of the paper. As a result, it was very difficult for this reviewer to follow the presentation and figure out what are the robust results and how they are related. I urge the authors to focus on the robust results in support of the major conclusions and delete unsubstantiated speculations. A streamlined, focused presentation is essential for this work to become publishable. Few would have the patience to finish reading an overreaching manuscript that mumbles along.

Major comments 1. Fig. 6c. Zonal-mean vertical velocity change does not seem robust judging from the vertical structure and the relationship with OLR. (a) Vertical velocity change around 10N, where the authors identified a decrease in zonal mean OLR, changes signs three times within the troposphere. The vertical structure is inconsistent with the first-baroclinic mode structure that dominates the tropical troposphere (e.g., Fig. 6d). (b) Strong upward anomalies between the equator and 5S are inconsistent with overall positive OLR anomalies in the region (Fig. 6a). Something seems wrong here. The same problem applies to the SVD mode showing a strong trend in Fig. 8.

The SVD mode with a strong trend features large OLR anomalies in the nonconvective subtropical regions (Figs. 8b,c), raising the question of to what extent the OLR pattern is associated with deep convection and deep vertical motion. Since the focus of the paper is on deep convection, I suggest using precipitation data throughout the paper, instead of OLR.

The discussion of the seasonal transition in climatology (Fig. 6, right) is out of blue and the physical relationship to the multi-decadal change in deep convection is ambiguous at the best. I suggest deleting the discussion of the seasonal transition to avoid confusion and streamline the paper.

2. The connection to the Brewer-Dobson circulation is dubious as it currently stands and should be deleted. ACP readers expect robustly tested results, not unsubstantiated

speculations. Figure 11 is bizarre: are the narrow bands in vertical velocity in Fig. 11a real, and the meridional dipole in the middle panel of Fig. 11b is averaged out in the meridional mean displayed in the upper panel of Fig. 11a. This is just one example that some speculations do not seem to hold water.

3. At the top of page 8, the authors admitted "a discussion of statistical significance . . . is practically impossible". This may be true but the mutual physical relationship among different fields is a minimum test an analysis needs to pass. My major comment 1 questions whether some of the results are robust and physically consistent.

4. Related to my major comment 1, JAS OLR change (Figs. 6 left, 8-9) is not zonally uniform; specifically the decrease in 10-20N is zonally confined in the African-western Pacific sector. This raises the question of whether the zonal mean even makes sense.

5. If the first three sections are hard to go through, section 4 (Summary and discussion) is impossible to comprehend, full of wild, poorly connected speculations. I urge the authors to summarize robust results that make physical sense first and only then, make some reasonable discussions that would be helpful for future research.

---

## Referee Comment (RC2) · Anonymous Referee #2 · 16 Apr 2018

The authors identified a northward shift of the ITCZ from observations and they related this shift to circulation changes in the stratosphere as a result of stratospheric cooling under global warming.

I like the fact that the authors tried a number of correlation analyses between the ITCZ shift and the changes in the tropospheric/stratospheric circulation. However, if I have not missed anything, I found some logical flaws in their arguments and that the mechanism proposed in Figure 13 is not rigorously supported by observations or theories. I wonder if it could be due to a problem of writing and presentation. Therefore, I recommend a major revision in the first revision round, so that the authors may clarify their

ideas, modify the title, write a more in-depth literature review, redo the data analysis, and remove unsupported claims.

The following critical comments may be considered during their revision.

1. The authors argued that changes in the equatorial stratospheric upwelling drove the ITCZ shift. This argument needs to be supported substantially:

(a) The movement of ITCZ under climate change has been well studied; see Schneider (GRL, 2017, https://doi.org/10.1002/2017GL075817) and references therein. These work have already demonstrated the importance of atmosphere-ocean exchange for determining the location of the ITCZ but the authors have not thoroughly review these observational and theoretical studies. (By the way, the authors mentioned the Hadley cell expansion in Introduction. They should clarify whether the Hadley cell expansion and the ITCZ movement are related.) Whatever the authors have come up with on explaining the ITCZ movements, their explanation (e.g. the stratospheric-led SST changes claimed in this paper) must eventually address those atmosphere-ocean exchange mechanistically and quantitatively.

(b) The logic that they turned their attention to stratosphere is a bit difficult to follow. They first pointed out that changes in convection extended into the tropical tropopause layer. Then they went further to assert that it was the changes in the stratospheric circulation that allowed the convection to get deeper into the tropical tropopause layer. However, there could be a lot of tropospheric causes (e.g. SST changes alone) that made the convection stronger and deeper regardless how the stratosphere changes. Without eliminating all those tropospheric causes first, it is hard to imagine why the stratosphere needs to be involved.

2. All proposed mechanism must be supported quantitatively but there is no detail how much fraction of the ITCZ movement may be caused by the proposed stratospheric changes. In fact, as a scientific paper, there is almost no number in the text that quantify any effects being studied. At most of the time, the authors present some correlation

coefficients as hints. But correlations or covariances cannot be used to imply causality between different variables. Causality should be at least shown by model simulations, which elucidates by how much changes in the stratospheric circulation would cause how much movements of the ITCZ. The models described on page 10 were not designed to isolate the stratospheric effects on the ITCZ movements. In addition, very little has been learned from those models: the model results actually do not quantitatively support their proposed mechanism shown on Figure 13. For this reason, the conclusion of the current manuscript that changes in the equatorial stratospheric upwelling drove the ITCZ shift appears to be speculative. Without a definitive modeling study, the proposed mechanism involving stratosphere, Section 3.4, and Figure 13 are deemed inappropriate.

3. Not only the causality should be quantified, the statistical significance of the observed quantities must also be established. Currently, there is no statistical test. Actually, the authors decided to skip the statistical test because "a discussion of the statistical significance of the relationships between variables that exhibit large trends in a short data record is practically impossible". This statement is hard to understand, because (i) if there were large trends in two time series, then it is almost certain that the correlation between these two time series is statistically significant because the noise is relatively small compared to the trends. Did the authors actually mean "weak trends"? And (ii) without establishing the statistical significance, the covariances of the stratospheric and tropospheric variables presented in the manuscript may just be artifacts.

4. A more serious problem perhaps is their use of the SVD analysis as a way to probe the mechanism. SVD is never intended for causality attribution. SVD can only extract the maximal variability from the data. The 1st mode of the correlation matrix can only be understood as the largest correlation in the tropospheric and stratospheric circulations that share the same secular trend over the past years. (Whether the secular trend is driven by anthropogenic warming is a separate question.) The SVD itself has no implication on whether the stratospheric circulation is driving the trospheric circulation, or vice versa. They may be independently responding to the same forcing (e.g. anthropogenic warming), and thus the covariance between the stratospheric and tropospheric trends can be almost certainly statistically significant, as discussed in Comment 3. But such trends do not help to "prove" the proposed mechanism.

5. When deriving the proposed mechanism, the authors correlated an index for stratospheric tropical upwelling ($I_\omega$) with tropospheric variables such as temperature, SST, and OLR. Figure 9a clearly showed that $I_\omega$ was a sum of a decreasing trend plus an oscillatory component. Were the correlation coefficients shown Figure 9b-9d caused by the trend or by the oscillatory component? (The maximal lag of 5 months shown in Figure 9e can only be related to the oscillatory component.) If it is caused by the trend, then, as Comment 4 suggested, the causality is not proven. Indeed, the authors may simply fit each of $\bar{\omega}$, $\bar{T}$, and OLR with a straight line $y = c_1 * time + c_2$ and plot $c_1$ to see whether similar Figure 9b-9d could be reproduced. If yes, then there is no direct evidence that $I_\omega$ is the cause of the tropospheric changes.

---

## Author Comment (AC1) · 3 Jun 2018

Few would have the patience to finish reading an overreaching manuscript that mumbles along.

Many thanks for the patience to read the manuscript and valuable comments.

Global climate change involves diverse aspects from the stratosphere to the ocean, from the polar region to the tropics, and from monsoon to severe storms. Each of these elements should be investigated independently in great detail, but their relationships to each other and their roles in global climate change also warrant investigation. Without the latter, we are unable to see the "big picture". The goal of this study is to provide a framework for putting some of the pieces together by elucidating the connection between the tropical atmosphere and ocean.

We explained this in the introduction of revised paper and also modified the title as "Role of tropical lower stratospheric cooling in deep convective activity".

**Major comments 1.**
Fig. 6c. Zonal-mean vertical velocity change does not seem robust judging from the vertical structure and the relationship with OLR. (a) Vertical velocity change around 10N, where the authors identified a decrease in zonal mean OLR, changes signs three times within the troposphere. The vertical structure is inconsistent with the first-baroclinic mode structure that dominates the tropical troposphere (e.g., Fig. 6d). (b) Strong upward anomalies between the equator and 5S are inconsistent with overall positive OLR anomalies in the region (Fig. 6a). Something seems wrong here.

Convective activity is largely different according to the geographical situation. In fact, "Strange feature of three times change of signs" in zonal mean vertical velocity around 10° N resulted from a superposition of different structures over the continental and oceanic sectors. The northward shift occurs mainly over the continental sector, but over the oceanic sector, strengthening of vertical velocity occurs around 5° N-10° N without latitudinal shift.

"Strong upward anomalies between the equator and 5° S", is due to a low level convergence over the western Pacific–New Guinea region. This is produced as a pair with the change over the eastern Pacific, a part of the Walker circulation. To facilitate to seeing the association between the change over the western and the eastern Pacific, original Figs. 5 and 10 were merged together and the zonal wind component south of the Equator was added in the revised version.

Then, you may ask why we need the zonal mean. Because, if the forcing is zonal, the lower tropospheric responses largely depend on the underlying geography with zonally asymmetric characteristics. However, above the tropical propose layer (TTL), the zonal mean component of the response becomes significant. Therefore, to investigate the variation related to the stratospheric changes, analysis of the zonal mean field is necessary.

To respond the question, the following text and Fig. A1 (Fig. 8 in revised paper) was added in revised version.

"Therefore, meridional sections of "standardized" mean JAS 1999- 2016 anomalous pressure vertical velocity were calculated for different sectors (Fig. 8). In the case of standardized

anomalies of 17-year mean as in Fig. 8, if one assumes a normal distribution, absolute values larger than 0.5 are statistically different from 0 at the 95% confidence level. Top panel shows the zonal mean field which can be comparable to that extracted by the SVD analysis in Fig. 7a. Contours indicate climatology. Middle panels are the same as the top panel, but divided into two parts: (left) African-Asian continental sector (30° W–130°E ) and (right) Pacific-Atlantic oceanic sector (130° E–330° E). Strengthening of the upward velocity in the TTL and the lower stratosphere occurs in the continental sector adding to the northward shift in the troposphere, but over the oceanic sector, strengthening of vertical velocity occurs around 5° N-10° N without latitudinal shift. In the oceanic sector, upwelling is largest around 5° N–10° N, and strengthening of the upwelling occurs at the same location as the climatology. If we limit the sector only over the African continent (20° W–20° E) to exclude Indian Ocean influence, the abovementioned continental characteristics becomes further clear (bottom left). Over the oceanic sector, increase of the vertical velocity generally occurs around 5° N (bottom- right), but in the western Pacific sector (130° E– 170° E), upward velocity mainly develops south of the equator (10° S– 0°) (bottom- middle). We also note that in climatological vertical velocity in the western Pacific sector is confined practically in the lower troposphere over the equatorial SH (10° S– 0°). This feature can be attributed to the fact that the convergence of the air occurs over a warm ocean east of New Guinea island (Fig. 3c).

The above results indicate that in spite of the mixture of different profile according sectors, the zonal mean vertical field in the TTL mainly reflects the variation over African-Asian continental sector.
.

[Figure]

Fig. A1

The same problem applies to the SVD mode showing a strong trend in Fig. 8. The SVD mode with a strong trend features large OLR anomalies in the nonconvective subtropical regions (Figs. 8b,c), raising the question of to what extent the OLR pattern is associated with deep convection and deep vertical motion. Since the focus of the paper is on deep convection, I suggest using precipitation data throughout the paper, instead of OLR.

The reviewer didn't indicate the exact location of "nonconvective subtropical regions". Relatively negative correlation is found in two regions higher than 20° N in original Fig. 8c, one around Pakistan, and the other over Western Pacific (reproduced in the top panel of Fig. A2). It should be noted that strong convection events occur in these region: severe floods in Pakistan in summer 2010 is well known. In the case of western Pacific east of Philippine, deep convection associated with the tropical cyclone frequently occurs (Fig. A2 bottom). Unfortunately, long-term datasets to study the global occurrence frequency of the extreme deep convection does not exist. We will show in Part II of this paper that following a decrease of tropical lower stratospheric temperature, extreme deep convection increased over the western Pacific in association with a development of tropical cyclones.

There are advantages and disadvantages of using precipitation data. Large amount of precipitation is produced by low level convergence, which is of little concern in the present study. We are only interested in precipitation associated with deep convection. Also, direct measurements of global precipitation have limited data record. Thus, precipitation is derived by making use of some assumptions. In this sense, the OLR is better.

[Figure]

Fig. A2 (top) heterogeneous correction map of OLR, (Bottom) Fraction of deep convective updraft in tropical cyclones (from Fig. 4 of Jiang and Tao, 2014).

Jiang, H. Y., and C. Tao, (2014) Contribution of tropical cyclones to global very deep convection. J. Climate,27, 4313–4336, doi:10.1175/JCLI-D-13-00085.1.

The discussion of the seasonal transition in climatology (Fig. 6, right) is out of blue and the physical relationship to the multi-decadal change in deep convection is ambiguous at the best. I suggest deleting the discussion of the seasonal transition to avoid confusion and streamline the paper.

We agree with the comment, and the discussion on the seasonal transition has been removed.

2. The connection to the Brewer-Dobson circulation is dubious as it currently stands and should be deleted. ACP readers expect robustly tested results, not unsubstantiated speculations.

In this paper we showed a connection between the zonal mean vertical velocity in the tropical lower stratosphere and in the TTL around the rising branch of the Hadley circulation. The stratospheric zonal mean meridional circulation can be designated as Brewer Dobson circulation.

Figure 11 is bizarre: are the narrow bands in vertical velocity in Fig. 11a real, and the meridional dipole in the middle panel of Fig. 11b is averaged out in the meridional mean displayed in the upper panel of Fig. 11a. This is just one example that some speculations do not seem to hold water.

It is difficult to verify whether the narrow bands are real because there are no available observations. However, it is plausible that these features are related to the vertical velocity perturbations near convective overshooting clouds (COV), which penetrate into the TTL beyond the level of neutral buoyancy. Figure A3-top panels show the same vertical velocity field in the original Fig. 11, but the mean anomalous vertical velocity of 1999-2016 is compared with climatological (2007-2017) occurrence frequencies of COV. Increased vertical velocity mainly occurs over the region where COVs are frequent. This relationship is expected because convective overshooting occurs in extreme deep convective clouds, which penetrate into the TTL.

[Figure]

Fig A3

In a previous paper we showed that daily variation of occurrence frequency of the COV during boreal winter is correlated with the zonal mean vertical velocity in the lower stratosphere, while that of the OLR is correlated with the vertical velocity in the troposphere as in Fig. A4.

[Figure]

Fig A4 (left) Correlation coefficient between the pressure coordinate vertical velocity at each pressure level and the daily convective overshooting occurrence frequency (COV) averaged over the tropics. (right) Same as the (left) but for the correlation with OLR. [from Kodera et al. (2015), The role of convective overshooting clouds in tropical stratosphere–troposphere dynamical coupling Atmos. Chem. Phys., 15, 6767-6774]

Similar relationship can also be found in climatological field over Africa. Figure A5 compares JAS climatology (2007- 2017 mean) between (left-hand panels) the vertical velocity at 500 hPa and OLR and (right-hand panels) vertical velocity at 100 hPa and the occurrence frequency of the COV over North Africa. At 500 hPa, upwelling extends zonally along 10 °N latitude corresponding to the region of low OLR, whereas upwelling at 100 hPa is broken up in a couple of segments, which roughly corresponds to the region of frequent COV. Similar relationship may also hold on a decadal timescale.

[Figure]

Fig A5

In recent paper, Taylor et al. (2017) reported an increasing trend in deep convective activity over Africa. They showed that the most intense Mesoscale Convective Systems (MCSs) with cloud top temperature lower than −70 °C (air temperature at ~150 hPa) exhibited the largest increasing trend; these MCSs represent clouds penetrating into the TTL. The evolution of these intense MCSs and the pressure vertical velocity at 150 hPa averaged over the similar area is compared in Fig. A6. Rectangles indicate the domains over which averages are calculated for the left panel. There is a quite good correspondence between the evolution of the vertical velocity at 150 hPa and the occurrence frequency of extreme deep convection over Sahel in Africa. It should be noted that the surface precipitation, and lower clouds do not show such clear increasing trend (Taylor et al., 2017). These evidence suggest that the vertical velocities shown in Fig. 11 appear to be related to perturbations induced by the distribution of extreme deep convection.

[Figure]

Fig A6

Top panels: (left) MCS frequency of which cloud top temperature lower than -70 °C. (right) Region of significant trends. The purple rectangle denote the domains used in the left panel. (from Taylor et al., 2017). (Bottom) JAS mean vertical velocity at 150 hPa averaged over the area shown by rectangular in right-hand panel.

Taylor, C. M. et al, 2017: Frequency of extreme Sahelian storms tripled since 1982 in satellite observations. Nature, 544, 475–478.

According to the comment, original Fig. 11 was replaced by Fig. A3 (Fig. 9 in revised paper) and the text was modified as follows in revised version.

" While bottom panels show a distribution of climatological (2007−2017) occurrence frequency of the convective over shooting (COV) in the same latitudinal zone. Inspection reveals that increasing trend of the upwelling occurs over the continental sector, especially where COVs are frequent. This characteristics are commonly seen in both summer hemispheres. The contrast between the continental and oceanic sectors is clearer in the SH where the distribution of lands is simpler. Because the COV occurs in the deep convective clouds penetrating into the TTL beyond the level of neutral buoyancy, such the increased vertical velocity in the TTL over the region of frequent COV seems reasonable. It should also be noted that a connection between the COV and the vertical velocity in the tropical lower stratosphere in day-to-day scale has been shown in a previous study by Kodera et al. (2015)."

3. At the top of page 8, the authors admitted "a discussion of statistical significance : : : is practically impossible". This may be true but the mutual physical relationship among different fields is a minimum test an analysis needs to pass. My major comment 1 questions whether some of the results are robust and physically consistent.

The results of above analyses can be schematically summarized in A7 (Fig. 11) . According to this, we selected 4 key variables which can be considered as fundamental in the recent tropical trends: (a) tropical lower stratospheric temperature in early summer (temperature at 70 h Pa averaged over 20° S–20° N from 16 July-16 August), (b) pressure vertical velocity at the bottom of the TTL (150 hPa) in August, (c) August-October mean "southward" winds south of the equator (10° S–0°) in the western hemisphere (180° W–0°) , (d) tendency in the SST, from the early summer (May-July) to late autumn (October–December) in the tropical Pacific west of South American continent (15° S–5° S, 100° W–80° W). Time series of these 4 variables (a–d) are displayed in A7 (Fig 11)–right.

When all 4 variables become negative (indicated by red dots), we define this as negative event. Similarly when all variables become positive (black dot), it is defined as positive event. All 6 positive events occurred within the first 14 years, while all 7 negative events appeared during the last 13 years. Chi-square test was conducted to examine whether such distribution of the events occurs by chance by dividing the whole 39 years to 3 equal period of 13-year. The result ($\chi^2 = 23$) indicates that such distribution occurs by chance with less than 0.1% of probability. Therefore, there is a statistically significant tendency that negative events occur more frequently in recent decade.

The problem here is, however, whether there is a causal relationship among the variables. We introduced a seasonal variation for the selection of the variable from the stratospheric cooling at the end of July, to a cooling of ocean from summer to autumn. The time evolution tentatively suggests a causality among them, which can be used as working hypothesis, but definite causality needs to be proven in future study. More detailed analysis on selected events will be done to understand causal relationship between the stratospheric and tropospheric change in Part II of this paper.

We added the above text and Fig. A7 (Fig. 11 in revised paper) to revised version.

[Figure]

Fig. A7

4. Related to my major comment 1, JAS OLR change (Figs. 6 left, 8-9) is not zonally uniform; specifically the decrease in 10-20N is zonally confined in the African-western Pacific sector. This raises the question of whether the zonal mean even makes sense.

Please see our response to comment 1.

5. If the first three sections are hard to go through, section 4 (Summary and discussion) is impossible to comprehend, full of wild, poorly connected speculations. I urge the authors to summarize robust results that make physical sense first and only then, make some reasonable discussions that would be helpful for future research.

As mentioned earlier in this paper, we have tried to provide a framework for examining the different regions and variables simultaneously to better understand the present global change occurring in the atmosphere-ocean system.

Causal relationship is usually verified by numerical model simulations. However, this requires a model that is capable of realistically reproducing all key processes, none seems existing today. Here, we focused on the validation of one feature of one model - the effects of extreme deep convection on the vertical velocity in the TTL - which is a key element in addressing the present problem.

According to comment, we simplified the discussion section.

---

## Author Comment (AC2) · 3 Jun 2018

We would like to thank you for the patience to read the manuscript and give valuable comments. We know it is not usual to investigate climate variation from the ocean surface to the stratosphere in a single paper, but we think it is necessary to get a global view of the recent change.

The authors identified a northward shift of the ITCZ from observations and they related this shift to circulation changes in the stratosphere as a result of stratospheric cooling under global warming. I like the fact that the authors tried a number of correlation analyses between the ITCZ shift and the changes in the tropospheric/stratospheric circulation. However, if I have not missed anything, I found some logical flaws in their arguments and that the mechanism proposed in Figure 13 is not rigorously supported by observations or theories. I wonder if it could be due to a problem of writing and presentation.

Therefore, I recommend a major revision in the first revision round, so that the authors may clarify their ideas, modify the title, write a more in-depth literature review, redo the data analysis, and remove unsupported claims. The following critical comments may be considered during their revision.

Global climate change involves diverse aspects from the stratosphere to the ocean, from the polar region to the tropics, and from monsoon to severe storms. Each of these elements should be investigated independently in great detail, but their relationships to each other and their roles in global climate change also warrant investigation. Without the latter, we are unable to see the "big picture". The goal of this study is to provide a framework for putting some of the pieces together by elucidating the connection between the tropical atmosphere and ocean.

We explain the above in introduction of the paper and also modified the title as Role of tropical lower stratospheric cooling in deep convective activity in revised version.

1. The authors argued that changes in the equatorial stratospheric upwelling drove the ITCZ shift. This argument needs to be supported substantially:

 (a) The movement of ITCZ under climate change has been well studied; see Schneider (GRL, 2017, https://doi.org/10.1002/2017GL075817) and references therein. These work have already demonstrated the importance of atmosphere-ocean exchange for determining the location of the ITCZ but the authors have not thoroughly review these observational and theoretical studies. (By the way, the authors mentioned the Hadley cell expansion in Introduction. They should clarify whether the Hadley cell expansion and the ITCZ movement are related.) Whatever the authors have come up with on explaining the ITCZ movements, their explanation (e.g. the stratospheric-led SST changes claimed in this paper) must eventually address those atmosphere-ocean exchange mechanistically and quantitatively.

Because the term ITCZ characterize only low level circulation, we do not use this term in the present study.

The characteristics of the convective zone over the ocean and continent largely differs. Figure A1 shows meridional sections of standardized JAS mean pressure vertical velocity (ω) in different sectors. Top panel is zonal mean ω, while middle panels show average ω by sectors: (left) continental (African-Asian), and (right) ocean (Pacific and Atlantic) sectors. Solid lines indicate climatological vertical velocities (-0.06, -0.03 Pa/s). We can see that the northward shift in the troposphere and large anomalous upwelling around the 100hPa level occur over the land sector. Over the oceanic sector, increase of vertical velocity is limited in the troposphere without changing latitudinal location. Therefore, we

consider the study of Schneider on the role of the interaction with the ocean is not so relevant to the present problem of northward shift of the continental convective zone.

Hadley cell expansion mentioned in the paper mainly concerns the descending branch of the Hadley cell as written in introduction, and whether that is related to tropical phenomena like the movement in ITCZ or convection is beyond the scope of this paper. This paper mainly concerns the ascending branch of the Hadley cell around the TTL, which exhibits also a different characteristic from the low level convergence zone.

To indicate the difference of vertical velocity field over the land and ocean sectors, original Fig. 6 was replaced by Fig. A1 (Fig. 8 in revised paper) and the following text were added in the revised version.

"Tropospheric zonal mean vertical velocity shows relatively small connection with the horizontal distribution of the OLR or SST in the SVD analysis in Fig. 7. This can be resulted from the fact that the regional scale variation dominates in the lower troposphere due to the surface geography.

Therefore, meridional sections of "standardized" mean JAS 1999- 2016 anomalous pressure vertical velocity were calculated for different sectors (Fig. 8). In the case of standardized anomalies of 17-year mean as in Fig. 8, if one assumes a normal distribution, absolute values larger than 0.5 are statistically different from 0 at the 95% confidence level. Top panel shows the zonal mean field which can be comparable to that extracted by the SVD analysis in Fig. 7a. Contours indicate climatology. Middle panels are the same as the top panel, but divided into two parts: (left) African-Asian continental sector (30° W–130° E ) and (right) Pacific-Atlantic oceanic sector (130° E–330° E). Strengthening of the upward velocity in the TTL and the lower stratosphere occurs in the continental sector adding to the northward shift in the troposphere, but over the oceanic sector, strengthening of vertical velocity occurs around 5° N-10° N without latitudinal shift. In the oceanic sector, upwelling is largest around 5° N–10° N, and strengthening of the upwelling occurs at the same location as the climatology. If we limit the sector only over the African continent (20° W–20° E) to exclude Indian Ocean influence, the abovementioned continental characteristics becomes further clear (bottom left). Over the oceanic sector, increase of the vertical velocity generally occurs around 5 °N (bottom–right), but in the western Pacific sector (130° E–170° E), upward velocity mainly develops south of the equator (10° S–0°) (bottom–middle). We also note that in climatological vertical velocity in the western Pacific sector is confined practically in the lower troposphere over the equatorial SH (10° S–0°). This feature can be attributed to the fact that the convergence of the air occurs over a warm ocean east of New Guinea island (Fig. 3c).

The above results indicate that in spite of the mixture of different profile according sectors, the zonal mean vertical field in the TTL mainly reflects the variation over African–Asian continental sector."

[Figure]

Fig. A1

(b) The logic that they turned their attention to stratosphere is a bit difficult to follow. They first pointed out that changes in convection extended into the tropical tropopause layer. Then they went further to assert that it was the changes in the stratospheric circulation that allowed the convection to get deeper into the tropical tropopause layer. However, there could be a lot of tropospheric causes (e.g. SST changes alone) that made the convection stronger and deeper regardless how the stratosphere changes. Without eliminating all those tropospheric causes first, it is hard to imagine why the stratosphere needs to be involved.

Convective activity depends not only the surface temperature but also that around the cloud top (level of neutral buoyancy). (see Emanuel et al., 2013). Therefore, it is natural to investigate the relationship between the change in deep convective activity and that of the TTL.

Figure A2 a and b are the same SVD analysis as in original Fig. 8b, except for the levels of vertical velocity: (a) 150- 50 hPa around the tropopause, and (b), 1000-300 hPa in the troposphere. It can be seen that the zonal mean ω in the troposphere around the rising branch of the Hadley circulation, 15° N-20° N is much closely related to the variation around the TTL than variation in the lower troposphere. According to the comment, original Figure 8 and related text have been replaced by Fig. A2 (Fig. 7 in revised paper) and the following text.

"For this, singular value decomposition (SVD) analysis (Kuroda, 1998) was conducted based on the normalized covariance (correlation) matrix between the zonal mean pressure

vertical velocity and the horizontal distribution of the OLR or SST in the tropics (30 °S–30 °N) during boreal summer (JAS) (Fig. 7). Each value at the grid point was weighted by the layer thickness in vertical, and the cosine of the latitude in meridional direction. To investigate the relative importance of zonal mean vertical velocity in different altitude, the SVD calculation are made with the zonal mean pressure vertical velocity of (a) 150–50 hPa and (b) 1000–300 hPa levels. To get a general view of the entire troposphere and stratosphere, the heterogeneous correlation of the vertical velocity is extended to a height range of 1000 to 10 hPa. The levels used for the SVD calculation are indicated by arrows.

In the case of the SVD with tropospheric vertical velocity (Fig. 7b), SVD 1 shows a variation related to the ENSO cycle, while that used the vertical velocity around the TTL (Fig. 7a) ENSO related variation appears as SVD 2. Here we show only those related to the decadal variation. Increasing trends in coefficients are evident in both cases. However, zonal mean ω around 15° N-20° N is much closely related to the variation around the TTL than variation in the lower troposphere. Accordingly, the variation of the OLR around 15° N over African-Asian sector is more closely related with the vertical velocity around the TTL in Fig. 7a. In the case of the SVD with tropospheric vertical velocity (Fig. 7b), heterogeneous correlation with the OLR shows the relationship with convective activity over the equatorial Pacific north of the equator which is confined mainly in the troposphere.

Therefore these results suggest a stronger connection between the convective activity around the rising branch of the Hadley circulation and lower stratospheric circulation through change in the TTL, similar to that suggested in transient response in previous works (Eguchi et al, 2015; and Kodera et al., 2015).

[Figure]

Fig. A2  Time coefficients and heterogeneous correlation maps from SVD analyses. Arrows indicate the level of vertical velocity used for the SDV calculation.

2. All proposed mechanism must be supported quantitatively but there is no detail how much fraction of the ITCZ movement may be caused by the proposed stratospheric changes. In fact, as a scientific paper, there is almost no number in the text that quantify any effects being studied. At most of the time, the authors present some correlation coefficients as hints. But correlations or covariances cannot be used to imply causality between different variables.

Causality should be at least shown by model simulations, which elucidates by how much changes in the stratospheric circulation would cause how much movements of the ITCZ. The models described on page 10 were not designed to isolate the stratospheric effects on the ITCZ movements. In addition, very little has been learned from those models: the model results actually do not quantitatively support their proposed mechanism shown on Figure 13. For this reason, the conclusion of the current manuscript that changes in the equatorial stratospheric upwelling drove the ITCZ shift appears to be speculative. Without a definitive modeling study, the proposed mechanism involving stratosphere, Section 3.4, and Figure 13 are deemed inappropriate.

We acknowledge that quantitative analysis is needed, but many change appear in recent decades in different regions, different seasons, and in different variables. Models can simulate only one aspect, and fail to reproduce changes globally. The goal of this paper is rather to provide a framework for testing these hypotheses in a numerical model, and information how the model should be improved for realistic simulation.

The shift of the convection should be resulted from nonlinear processes. Therefore it may not be possible to linearly separate the portion of the stratospheric influence.

Models are a powerful tool for demonstrating causal relationships, if they are able to reproduce all key processes. In the present case, the model must be able to successfully simulate the effects of extreme deep convection penetrating to the TTL. In this paper, we show that (original Fig. 12) the JMA model has difficulty simulating the vertical velocity in the TTL over land without assimilation of the observational data. This appears to be a common limitation in current models. Until the models are improved, realistic numerical simulation of the decadal variation of extreme deep convective activity discussed in this study may be difficult.

3. Not only the causality should be quantified, the statistical significance of the observed quantities must also be established. Currently, there is no statistical test. Actually, the authors decided to skip the statistical test because "a discussion of the statistical significance of the relationships between variables that exhibit large trends in a short data record is practically impossible". This statement is hard to understand, because (i) if there were large trends in two time series, then it is almost certain that the correlation between these two time series is statistically significant because the noise is relatively small compared to the trends. Did the authors actually mean "weak trends"? And (ii) without establishing the statistical significance, the covariance of the stratospheric and tropospheric variables presented in the manuscript may just be artifacts.

We will make use of the standardized anomalies to demonstrate the statistical significance of the change in vertical velocity: Standardized anomalies of 17-year mean pressure vertical velocity is shown in Fig. A1. In this case, if one assumes a normal distribution, absolute values larger than 0.5 are statistically different from 0 at the 95% significance level. We can also see that the largest difference appears around the tropopause region in the continental sectors.

See also our response to the next question 4.

4. A more serious problem perhaps is their use of the SVD analysis as a way to probe the mechanism. SVD is never intended for causality attribution. SVD can only extract the maximal variability from the data. The 1st mode of the correlation matrix can only be understood as the largest correlation in the tropospheric and stratospheric circulations that share the same secular trend over the past years. (Whether the secular trend is driven by anthropogenic warming is a separate question.) The SVD itself has no implication on whether the stratospheric circulation is driving the tropospheric circulation, or vice versa. They may be independently responding to the same forcing (e.g. anthropogenic warming), and thus the covariance between the

stratospheric and tropospheric trends can be almost certainly statistically significant, as discussed in Comment 3. But such trends do not help to "prove" the proposed mechanism.

Please be noted that we do not try to demonstrate any causality with the SVD: we only use the SVD analysis to extract covariance between two fields and its spatial structures as you mentioned. In revised version, we compare the results of two SVD analysis using different height of the vertical velocity field. This permitted to study which region of vertical velocity is more closely related to the recent trend on the surface.

5. When deriving the proposed mechanism, the authors correlated an index for stratospheric tropical upwelling ($I\omega$) with tropospheric variables such as temperature, SST, and OLR. Figure 9a clearly showed that $I\omega$ was a sum of a decreasing trend plus an oscillatory component. Were the correlation coefficients shown Figure 9b-9d caused by the trend or by the oscillatory component? (The maximal lag of 5 months shown in Figure 9e can only be related to the oscillatory component.) If it is caused by the trend, then, as Comment 4 suggested, the causality is not proven. Indeed, the authors may simply fit each of $\bar{\omega}, \bar{T}$, and OLR with a straight line y = c1 * time + c2 and plot c1 to see whether similar Figure 9b-9d could be reproduced. If yes, then there evidence that is the cause of the tropospheric changes.

Generally, year-to-year variation contains variability produced by different sources from that producing decadal trend. For instance, interannual variation in the stratospheric contains stratospheric QBO signal, while tropospheric variation includes ENSO signal. Therefore the correlation study of rapid component by eliminating linear trend may not help for understanding the causal relationship among the trends.

The results of above analyses can be schematically summarized in Fig. 11-left. According to this, we selected 4 key variables which can be considered as fundamental in the recent tropical trends: (a) tropical lower stratospheric temperature in early summer (temperature at 70 h Pa averaged over 20° S–20° N from 16 July–16 August), (b) pressure vertical velocity at the bottom of the TTL (150 hPa) in August, (c) August–October mean "southward" winds south of the equator (10° S–0°) in the western hemisphere (180° W–0°), (d) tendency in the SST, from the early summer (May–July) to late autumn (October–December) in the tropical Pacific west of South American continent (15° S–5° S, 100° W–80° W). Time series of these 4 variables (a–d) are displayed in Fig 11–right.

When all 4 variables become negative (indicated by red dots), we define this as negative event. Similarly when all variables become positive (black dot), it is defined as positive event. All 6 positive events occurred within the first 14 years, while all 7 negative events appeared during the last 13 years. Chi square test was conducted to examine whether such distribution of the events occurs by chance by dividing the whole 39 years to 3 equal period of 13-year. The result ($\chi^2 = 23$) indicates that such distribution occurs by chance with less than 0.1% of probability. Therefore, there is a statistically significant tendency that negative events occur more frequently in recent decade.

The problem here is, however, whether there is a causal relationship among the variables. We introduced a seasonal variation for the selection of the variable from the stratospheric cooling at the end of July, to a cooling of ocean from summer to autumn. The time evolution tentatively suggests a causality among them, which can be used as working hypothesis, but definite causality needs to be proven in future study. More detailed

analysis on selected events will be done to understand causal relationship between the stratospheric and tropospheric change in Part II of this paper.

We added the Fig. A3 and above text to revised version.

[Figure]

Fig A3.

---

## Author Response (AR4)

Many thanks for your help for improving the manuscript. We corrected or modified the text according to the comments.

Comments to the Author:
The paper is now very close to a form that I will feel comfortable in accepting for ACP. Please can you consider the further detailed comments below, which are primarily concerned with wording, and make appropriate changes. I will then accept the paper.

**p1 l16**: 'resulted from' > 'were associated with' [There is no clear evidence that the Hadley Circulation is dynamically separate from the 'cross-equatorial southerlies' and therefore that a change in the Hadley Circulation is something that can be regarded as a CAUSE of a change in the cross-equatorial southerlies.]

    The phrase 'resulted from' was changed to 'were associated with'

**p1 l27**: 'are noted' doesn't seem necessary -- is it a relic from a previous edit?

    Removed

**p2 l8:** '(Emanuel et al 2013)'

    Corrected as '(Emanuel et al., 2013)'

**p2 l32**: 'fundamental cause is not ... but rather a strengthening of the deep ascending branch of the summertime Hadley circulation extending into the stratosphere' -- again in what sense can the strengthening of the Hadley circulation be regarded as a cause -- i.e. in what sense is it dynamically separate from the aspects of the tropical tropospheric circulation that you are regarding as changing? Elsewhere you have identified increase in land surface and changes in lower stratospheric temperatures as possible causes -- that potentially makes more sense to me.

    According to the comment, the sentence was modified as follows.
'In this paper, we suggest that the fundamental driver of the recent decadal trend in the tropics from around 1999 is a strengthening of the deep ascending branch of the summertime Hadley circulation associated with a cooling in the lower stratosphere and a warming in the troposphere.'

**p5 l1**: second quotation mark missing on 'deep ascending branch'

    The quotation mark added.

**p10 l11**: '(a) Cooling in the lower stratosphere adds to the global warming in the troposphere.' -- this sounds as if cooling in the lower stratosphere REINFORCES warming in the troposphere. Do you really mean that? Are you simply saying that the result of increase in greenhouse gases is warming in the troposphere and cooling in the stratosphere?

    The sentence was changed as follows.
'(a) Cooling of the lower stratosphere due to the direct radiative effect and dynamic effect of the stratospheric mean meridional circulation. The following sentences were added.'

**p11 l19**: Bearing in mind the referee comments, personally I think it would be much better to add a sentence here along the lines of: 'However, suitably designed numerical experiments will be needed to add further support to these ideas.'

The following sentences were added.

'Current global models have a difficulty to accurately simulate the effect of extreme deep convection on the TTL. Suitably designed numerical experiments using global models with improved convective parameterizations will be needed to add further support to these ideas.'

**p12 l1:** 'It should also be noted that the expansion rates of the tropics should be much smaller than those reported in past studies (Staten et al., 2018).' -- 'should' isn't the correct word to use here. A possible modification would be 'are much smaller' -- but that isn't really consistent with what Staten et al (2018) are saying. In their abstract they say 'However, although theory and modelling suggest increasing GHG concentrations should widen the tropics, previous observational-based studies depict disparate rates of expansion, including many that are far higher than those simulated by climate models.' -- i.e. some estimates seem much too large, but not that all estimates seem much too large. The simplest thing would be to omit this sentence -- but you could simply say something like 'The broad range of estimates for tropical widening has recently been discussed by Staten et al (2018).'

The sentence was removed.

References: The Evan and Camargo (2011) is still in the reference list, though you have removed the citation from the text. Please check general consistency between citations in text and reference list.

Removed from the list.

[revised manuscript text omitted]